# Perceptual decisions and oculomotor responses rely on temporally distinct streams of evidence

Matteo Lisi [1,2,3✉], Michael J. Morgan[1] & Joshua A. Solomon [1✉]

Perceptual decisions often require the integration of noisy sensory evidence over time. This process is formalized with sequential sampling models, where evidence is accumulated up to a decision threshold before a choice is made. Although intuition suggests that decision formation must precede the preparation of a motor response (i.e., the action used to communicate the choice), neurophysiological findings have suggested that these two processes might be one and the same. To test this idea, we developed a reverse-correlation protocol in which the visual stimuli that influence decisions can be distinguished from those guiding motor responses. In three experiments, we found that the temporal weighting function of oculomotor responses did not overlap with the relatively early weighting function of stimulus properties having an impact on decision formation. These results support a timeline in which perceptual decisions are formed, at least in part, prior to the preparation of a motor response.

[1] Centre for Applied Vision Research, City, University of London, London, UK. [2] Department of Psychology, University of Essex, Colchester, UK. [3] Department of Psychology, Royal Holloway, University of London, Egham, UK. ✉email: matteo.lisi@rhul.ac.uk; j.a.solomon@city.ac.uk

When making quick decisions about uncertain sensory stimuli, humans and other animals display speed-accuracy trade-offs, indicating that they can accumulate information over time to improve their performance. Indeed, computational models positing the accumulation of evidence up to a decision threshold can account for both response time and accuracy in many forced-choice tasks[1]. Beyond their usefulness for analyzing and interpreting behavioral data, such models have important implications for the neural mechanisms underlying the decision process[2]. For instance, in addition to neurons that encode the instantaneous sensory evidence, many models postulate the existence of neural accumulators, i.e., neurons or neuronal populations that perform the integration of sensory evidence over time[2,3]. While the instantaneous evidence is thought to be represented in sensory processing areas, such as in the middle temporal area (MT) for motion stimuli, correlates of evidence accumulation have been identified in frontal[4] and parietal areas[5,6] of primate brain, which are also involved in the planning and execution of the motor response used to communicate the decision.

The finding of neural correlates of evidence accumulation in motor and pre-motor areas has led to the influential intentional framework[7], according to which, 'perceptual decision-making is implemented in the brain as a process of choosing between available motor actions rather than as a process of representing the properties of the sensory stimulus'[8]. This implies that during decision-making there would be a continuous flow of information from sensory to motor areas, producing graded levels of readiness to execute motor responses that are proportional to the time integral of the sensory evidence. Support for the intentional framework comes from neurophysiological investigations of the lateral intraparietal area (LIP), which seems to implement both sensorimotor transformation for eye movements (e.g., ref. [9]) as well as accumulation of sensory evidence in tasks that require oculomotor responses[5,8]. Nonetheless, there are a number of caveats to bear in mind. While some neurons in LIP may represent accumulated evidence for some perceptual tasks, it is clear that other neurons in LIP represent (unaccumulated) instantaneous evidence for other tasks[10] Furthermore, motion discrimination was found to be unimpeded by reversible inactivation of LIP[11], despite LIP neurons' clear accumulator-like properties in this task. Reconciling these seemingly inconsistent results are suggestions of independence between decision-related and oculomotor signals in LIP[12] and multiplexing of decision-related and decision-irrelevant signals in LIP[13–15].

In order to refine our understanding of information processing during speeded perceptual decisions, we designed a reverse-correlation paradigm in which the time course of a perceptual decision can be disentangled from the time course of saccadic preparation. Key to our paradigm is the stochastic resampling of stimulus properties, including the saccadic target position, during decision formation (at a rate of 15 samples per second). If decision-making were implemented in the brain as a process of choosing between motor actions, then we would expect the relative impact of each interval on the perceptual decision to be similar to the relative impact of each interval on the saccadic endpoint. However, to anticipate our results, across 3 experiments we find that the temporal weighting functions for decision-formation and motor-preparation were distinct and largely non-overlapping. Whereas early samples predicted the perceptual decision, later samples (approximately 200 to 50 ms before the onset of the saccade) predicted the parameters of the saccade but not the decision. Since our results are based only on behavioral measurements, they do not speak against the idea that the motor system implements the accumulation of sensory evidence. However, by showing that the precise parameters of the eye movement

were determined by information sampled after the decision, our results demonstrate that the motor response was not yet ready to launch when the decision process terminated. This suggests that perceptual decisions and speeded, oculomotor responses rely on temporally distinct streams of evidence.

## Results

In Experiment 1, human observers were presented with two peripheral targets (Fig. 1a), whose positions were re-sampled at 15 Hz from two generative distributions, and asked to decide with of the two distributions was closer to the central fixation point. To identify the timing of visual influences on decision-making and motor planning, we aligned noisy position samples with respect to the saccadic onset time and correlated them with either the binary choice (left vs. right target) or to the endpoint of the saccadic eye movement. This allowed purely temporal characterizations of the target position's influence, not only upon the choice (i.e., the saccade's direction: right or left; black trace in Fig. 1b) but its eventual endpoint as well (blue trace in Fig. 1b). We estimated the evolution of these effects as a function of the temporal distance from saccade onset by using a Bayesian approach to reverse correlation (see "Methods" for details). This analysis allowed us to reconstruct the temporal weighting functions underlying the decision and the oculomotor response. The results (Fig. 1e) revealed temporal weighting functions that were distinct and largely non-overlapping; whereas choices were correlated only with relatively early samples, saccadic endpoints were correlated with later samples. Our analysis thus revealed that decision-formation and motor-preparation in Experiment 1 accrued the visual input that guided them over distinct temporal intervals.

The display of Experiment 1 differed from that of the most common paradigms used in the literature because it required monitoring two peripheral locations instead of a single, more central location (see "Discussion" for the caveats that apply in the interpretation of Experiment 1). The results thus leave open the possibility that integration of visual evidence could proceed in parallel under different conditions where the perceptual decision does not involve judgments about the peripheral saccadic targets. This possibility was addressed with Experiment 2, in which two patches of varying luminance were presented to opposite edges of the fovea. The perceptual decision involved choosing which was brighter on average (see Fig. 1c). We estimated the temporal weighting functions using the same approach as in Experiment 1 and found again distinct, largely non-overlapping temporal weighting functions (Fig. 1f). The results of Experiment 2 thus indicate that visual input does not inform simultaneously decision formation and the preparation of the motor response, regardless of the particular visual feature that needs to be processed for the perceptual decision (position vs. brightness) or on the location of the visual signals (peripheral vs. parafoveal).

One possible concern for the interpretation of the results obtained in Experiments 1 and 2 is that while integrating visual information over time is required by the perceptual decision task, it is not required by the saccadic task—observers could have simply made a saccade toward the last target position that they registered on the side that they had chosen. According to this account, the difference we observe in the temporal weighting function would be due to different task demands. Although this explanation cannot fully account for the pattern seen in Experiments 1 and 2 (see "Discussion"), we ran an additional experiment to test this directly. In Experiment 3, observers were explicitly instructed to shift their gaze toward the mean of target locations (the centroid) and were given trial-by-trial feedback on the accuracy of their saccades (see Figure S1b). We also increased

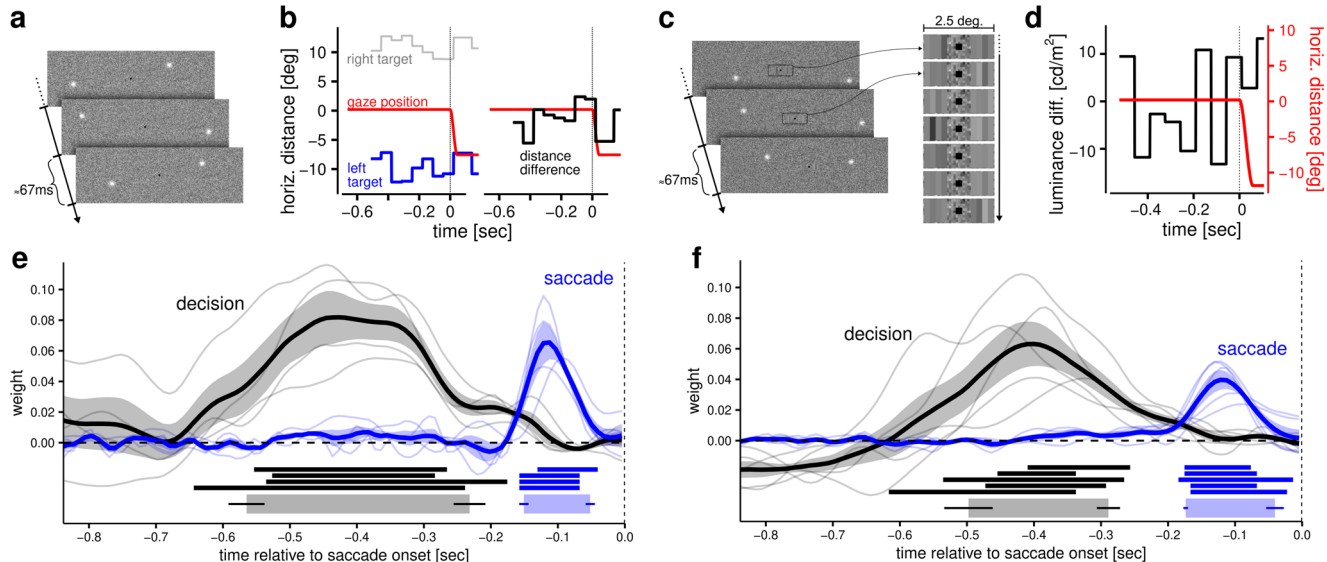

**Fig. 1 Experimental protocols and estimated temporal weighting functions.** In Experiment 1 (**a**) observers began each trial by looking at a central fixation dot, then two peripheral luminance targets appeared, with their positions changing at 15 Hz. The horizontal positions of the two targets in one trial are plotted as a function of time in the left facet of panel **b**. These distances form the basis of decisions regarding which of the two targets was closer to the fixation point. Their differences are plotted in the right facet of panel **b**. Observers were required to report their decisions by looking at the closer target, and precise estimates for saccadic onset times were obtained offline from gaze recordings. Vertical dashed lines in panels **b**, **d**, **e**, and **f** indicate saccade onset. In Experiment 2 (**c**), the observers were required to judge which of two patches, composed of four vertical bars and presented at the edge of the fovea, had the greatest average luminance. Panel **d** shows the average luminance difference as a function of time in one example trial. In this case, the two distributions from which target positions were sampled always had the same distance from fixation (10 deg) and observers were asked to look, as quickly as possible, at the target placed on the side of the brightest patch. Reverse-correlation analyses revealed temporally distinct weighting functions for perceptual decisions and oculomotor responses (Experiment 1, panel **e**; Experiment 2, panel **f**). The weighting functions reveal the influence of visual information on perceptual decisions (black line) and saccade planning (blue lines). (Note that although the weights for saccade and decision have similar magnitude, they are expressed in different units.) Thin lines represent weighting functions of individual observers, thick lines represent the group average, and the error bands represent the standard error of the mean across observers. Underneath the curves, the horizontal straight lines represent intervals (integration windows) in which the estimated weights were different from zero (each line representing a participant). The thicker horizontal bars at the bottom represent the average integration windows, obtained by averaging the onset and offset of the integration window for each participant (see "Methods" for details). The horizontal thin lines represent the bootstrapped standard errors on the onset and offset of the group-level integration window.

the positional uncertainty of the centroids, by introducing additional random variations relative to the fixation (see "Methods" for detail). This was done to motivate observers to estimate the centroids on a trial-by-trial basis, rather than simply making a saccade to stereotypical locations to the left and right of the fixation point. Despite the additional uncertainty induced by this manipulation, observers achieved a similar or even slightly smaller average saccadic error than in Experiment 1. The mean saccadic error (relative to the centroid) in Experiment 3 was 2.07 deg (SD 0.34), while it was 2.44 (SD 0.76) in Experiment 1. A comparison of saccadic variability in this experiment with previous data that used the same target[16] supported the notion that observers averaged more than 1 position sample to direct their saccades (see Figure S2 for details). Importantly, despite the differences in design and task instructions, the results of Experiment 3 fully replicate the pattern seen in the previous experiments (see Figure S1a).

Finally, we assessed whether the total duration of the pre-saccadic interval influenced the overlap of the temporal weighting functions. We split trials (pooling data from all three experiments) into 4 bins according to individual quartiles of saccadic latency and estimated weighting functions separately for each latency bin (Fig. 2). Across the 4 bins we found a relationship between speed and accuracy (Fig. 3), whereby slow responses were less likely to be accurate. This is likely due to trial-by-trial fluctuations in decision difficulty due to the staircase procedure. Most interestingly, we found a very similar pattern with little/no overlap between weighting functions in each latency bin,

including those with faster responses. Thus, the dissociation between the accrual of information for a perceptual decision and the accrual of information for motor planning is robust and present even when the decision and motor preparation unfold over a very short time (mean latency in the fastest bin was 473 ms; SD across participants, 76 ms).

## Discussion

We investigated whether sensory information accrued during speeded perceptual decisions could simultaneously inform the decision process and the motor preparation of a saccadic response used to communicate the choice. We designed an experimental protocol in which both the evidence for the perceptual decision as well as the position of the saccadic target was varied over time. By using reverse-correlation on both choices and saccadic endpoints, we identified the temporal integration windows of decision formation and motor preparation and found that they were largely distinct and non-overlapping. The different time courses of visual influence on decision formation and eye-movement preparation point to distinct accruals of sensory information for these two processes, possibly having a serial or cascaded organization.

In Experiment 1, participants were asked to judge which of the 2 peripheral targets was on average closer to the central fixation point. Although the results suggested that perceptual decisions and oculomotor responses were supported by distinct accruals of information, it is possible that the specific characteristics of the paradigm used in Experiment 1 encouraged a serial strategy that

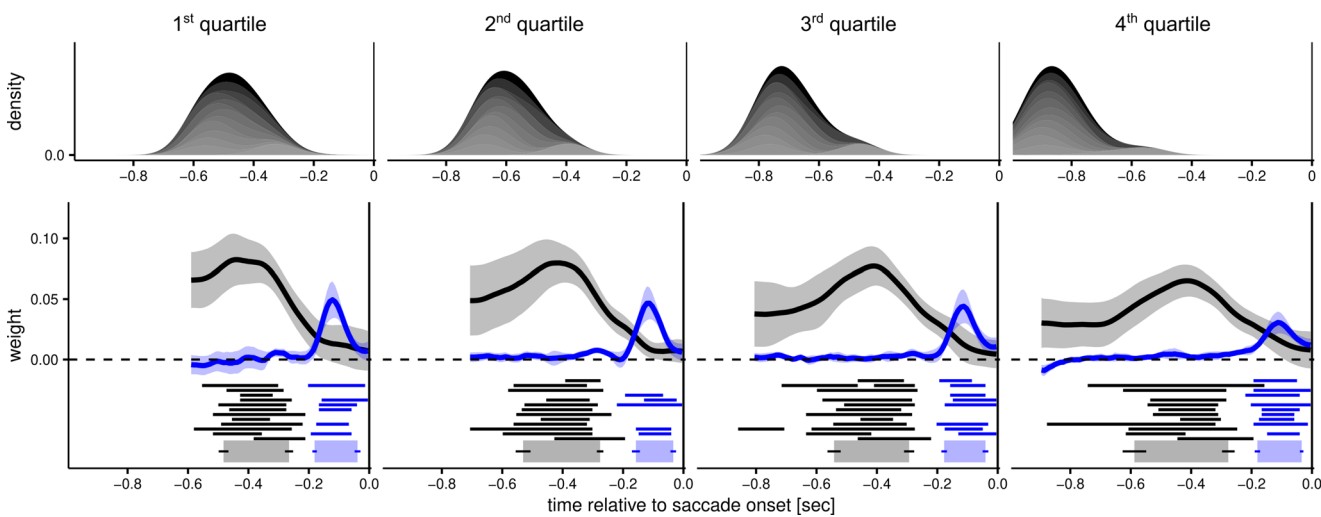

**Fig. 2 Temporal weighting functions and saccadic latency.** This figure represents temporal weighting functions as a function of saccadic latency. The latency of the responses is represented in the upper panels as the time of target onset relative to the saccade onset, thus slower trials (right-hand panels) show distributions centered on more negative values. To estimate these functions, we pooled together data from Experiments 1, 2, and 3, and split the data according to the quartiles of individual distributions of saccadic latency. Different shades of gray represent different participants. The lower panels represent the weighting functions with the same conventions used in Fig. 1. Note that for some participants, in some latency bins, the horizontal line representing the integration windows are lacking. These indicate cases in which, after binning the data, there was not enough information to determine reliably the integration windows, as revealed by broad 95% Bayesian credible intervals that encompassed zero at all time points. These cases have been excluded from the calculation of the group-level integration windows (bottom, thick lines), but were nevertheless included in the calculation of the average weighting functions.

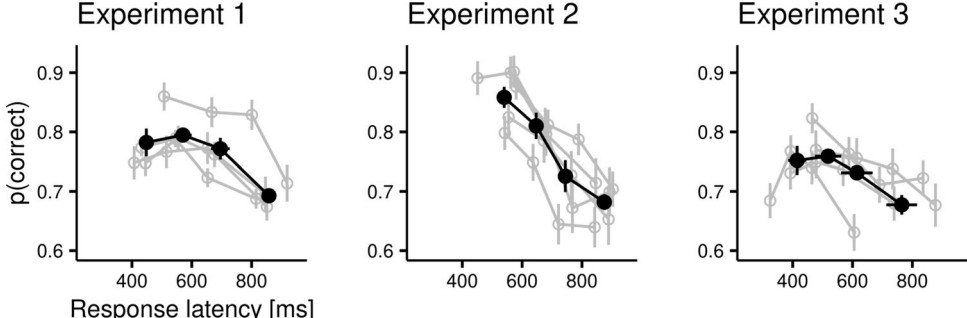

**Fig. 3 Relationship between speed and accuracy.** Across the 3 experiments, participants displayed a negative correlation between the accuracy (on the vertical axis) and the latency of the responses (horizontal axis). In the figure, gray dots and lines represent individual participants, and black lines the group averages. Data have been binned according to quartiles of individual latency distributions. All error bars are bootstrapped standard errors.

resulted in non-overlapping weighting functions. While the perceptual decision required observers to compute the difference in distance from fixation between the two targets, the oculomotor response required only the gaze-centered coordinates of the chosen target. Thus, is possible that the requirement of computing the difference in position interfered with the processing of the gaze-centered coordinates, forcing observers to program an appropriate eye movement only after having selected the appropriate target. Moreover, using a dual-task manipulation, a previous study showed different time courses for peripheral and foveal processing of visual information before an eye movement: while peripheral processing stopped 60–80 ms before the saccade was launched, foveal processing continued until saccade onset[17]. Accordingly, when decoupling decision-relevant stimulus properties from the saccadic targets in Experiment 2, we ensured that the former would appear close to the fovea (see Fig. 1c). Nonetheless, the temporal weighting functions collected in Experiment 2 were similar to those in Experiment 1. Consequently, the most straightforward summary of our results is that, in both experiments, sensory information received at any point in time

contributed to either the formation of the decision or the preparation of the motor response. Although we cannot exclude the possibility that relatively early samples (>200 ms before the saccade) may have suggested spatially imprecise motor plans, our results unequivocally demonstrate that the precise coordinates of saccadic endpoints were determined by later samples, which did not contribute to the formation of the perceptual decisions.

One possible issue that was not addressed in Experiments 1 and 2 concerned task instructions. If saccadic programming occurred simultaneously with the integration of perceptual evidence, then we would naturally expect some degree of overlap between saccadic and perceptual weighting functions. However, it is not clear how much overlap we should expect, given that observers were not discouraged from aiming their saccades toward a single target position. To address this issue, we designed Experiment 3, in which observers were instructed explicitly to shift their gaze to the centroid (the mean of the blob's distribution). Observers received trial-by-trial feedback on the accuracy of their saccades (see "Methods" for details). Nonetheless, the results of this experiment fully replicated the pattern seen in

Experiments 1 and 2, thus corroborating an interpretation of our findings in terms of temporally distinct streams of evidence for perceptual decisions and oculomotor responses.

An important question is whether our findings can be generalized to other conditions, such as free viewing of stable scenes, where fixation durations are on the order of just 300 ms (ref. [18]), which is considerably less than the sum of integration times for perceptual decision-making and saccade planning in our paradigm. Of course, this question cannot be answered using our paradigm, which depends on stochastic resampling of stimulus parameters. However, in stable scenes, visual information can be retained and combined across multiple fixations, an idea supported by many lines of research. For example, it has been shown that the influence of visual information accumulated during a fixation is not limited to the first saccade following the fixation but extends to subsequent saccades[19]. Other studies have demonstrated that visual information can be integrated across saccades in a near-optimal fashion[20,21], and that attention can be allocated stably across eye movements in the presence of visual landmark[22]. Thus, when free-viewing stable scenes, the accumulation of perceptual evidence required to inform upcoming decisions and motor actions does not need to be completed within a single fixation; it may extend across multiple fixations. In contrast, in our experiments, the accumulation of evidence had to start anew at each trial and the difficulty of perceptual decisions was set to elicit a substantial proportion of errors, resulting in relatively slow response times and long integration windows.

We note that the temporal weighting functions for saccadic eye movements derived from our data are fully consistent with previous measurements, and they replicate critical features already reported in the literature, such as the presence of a saccadic dead time; a "point of no return" after which afferent information is too late to influence the upcoming movement[23,24]. Moreover, the direction of the influence of target-position samples on the saccadic landing was always positive (i.e., saccadic landing positions were attracted toward each sample, not repelled away), consistent with the integration of position information and inconsistent with repulsion by distractors, which usually occurs for saccade latencies longer than 200 ms (ref. [25]). Our results also reveal that despite the relatively long presentation of the stimuli, the saccadic system integrates information over only a relatively narrow temporal window ($\approx$100 ms). Similarly, narrow windows were found in studies of saccades to moving targets[26–28].

As mentioned in the Introduction, a strict interpretation of the intentional framework[7,29] would predict largely overlapping temporal weighting functions for decision formation and motor preparation. Recent studies, however, have questioned whether motor areas in the brain actually do play a central role in evidence accumulation[13,15]. Our results contribute to this debate by showing that visual input does not seem to simultaneously inform the formation of a perceptual decision and the preparation of a saccadic response. This perspective is in line with a recent study[30] of economic (value-based) decision-making, which found evidence for sequential encoding of choice and action preparation in the macaque brain. Specifically, neurons in the supplementary eye fields (SEF) were found to encode first the value of the chosen option and—about 100 ms later—the parameter of the saccadic response that would obtain it. Although this study did not use time-varying visual stimuli, and therefore did not involve accumulation of visual signals, it nevertheless points to a sequential organization of decision-formation and response-preparation that may apply also to non-economic decisions. Indeed, such a sequential organization would be fully consistent with the temporal weighting functions estimated in our study.

Although our results challenge the idea that oculomotor responses are prepared in parallel with the accumulation of perceptual evidence, they do not address the question of whether other types of responses (e.g., manual) can be prepared concurrently with decision formation. Indeed, unlike saccades, hand movements can be modified online in response to new sensory inputs and often respond differently to stimuli or tasks that require the integration of information over time[31,32]. Indeed, one previous study using motor perturbations[33] found evidence for a continuous flow of information from the ongoing decision process to control system for hand movements in the brain. In that study motor activity gradually built up with a rate that (averaged over trials) depended on the evidence discriminability.

In summary, our results demonstrate that, in a speeded perceptual decision task, the integration of visual signals for planning oculomotor responses terminates later than the accumulation of evidence that inform the perceptual decision. These results are suggestive of a serial organization of evidence accumulation and motor preparation, and they are in line with theoretical models developed to account for psychological effects such as the refractory period and attentional blink[34,35], which hypothesize a temporal separation of these processes. Such theories postulate the existence of central bottlenecks to explain why, despite its massively parallel architecture, the brain can be surprisingly slow and serial at performing certain tasks. Indeed, a recent study provided evidence for a bottleneck that prevents incorporating evidence for multiple decisions in parallel[36]. Sequential and dissociable processes for evidence accumulation and motor preparation may even facilitate the re-calibration of behavioral responses in changing environments. Although this strategy might carry costs, such as slower response times, the benefits coming from the increased flexibility may outweigh the costs.

## Methods

**Participants.** Four observers (2 authors and 2 naive observers) participated in Experiment 1; 5 observers participated in Experiment 2 (1 author and 4 naive observers); finally, 1 author and 3 naive observers participated in Experiment 3. These sample sizes were chosen in accordance with similar psychophysical reverse correlation studies in the literature (see ref. [37] for a review). All had normal or corrected-to-normal vision. Participants gave their informed consent in written form; the protocol of the study received full approval from the Research Ethics Committee of the School of Health Sciences of City, University of London.

**Apparatus.** The experiments were run in a quiet, dark room. Right eye gaze position was recorded with a video-based eye tracker (Eyelink 1000, SR Research Ltd., Mississauga, Ontario, Canada). The participant's head was placed on a chinrest with adjustable forehead rest. Visual stimuli were presented on a gamma-linearized LCD monitor, 0.515 m wide, at a viewing distance of 0.77 m. The monitor resolution was 1920 × 1200. An Apple computer controlled stimulus presentations and response collection. The experimental protocol was implemented using MATLAB (The MathWorks Inc., Natick, Massachusetts, USA) and the Psychophysics[38] and Eyelink[39] toolboxes.

**Stimuli.** Stimuli were 2-D blobs with a Gaussian luminance profile presented on a background made of squares (side $\approx$ 0.08 deg), with random luminance drawn from a Gaussian distribution (RMS contrast $\approx$10%). The space constant of each blob was set to 0.3 deg and their peak luminance was $\approx$147 cd/m$^2$. The position of each blob kept changing at 15 Hz (every 4 monitor refresh cycles, corresponding to 67 ms) and was drawn randomly from a 2-D uniform distribution. The size of the distribution was adjusted so that the standard deviation of position samples was 1.5 deg (thus yielding distances from the mean up to 2.6 deg). In addition to the peripheral Gaussian blobs, Experiment 2 included also two small squares presented near fixation (side $\approx$ 0.8 deg, centered at $\approx$0.8 deg to the left and right side of the fixation point). Each square was divided into 4 vertical bars, and the luminance of each bar kept changing at 15 Hz, from a Gaussian distribution with standard deviation of 10 cd/m$^2$ and mean equal either to the mean background luminance ($\approx$46 cd/m$^2$) or to a higher value set according to a staircase procedure (details in the "Procedure" section).

## Procedure

*Experiment 1.* In our protocol, observers were asked to make a speeded discrimination and to report their choice by means of a saccadic eye movement. In Experiment 1 they were presented with two peripheral targets (Fig. 1a), whose positions were re-sampled at 15 Hz from two generative distributions (see "Stimuli"

section) and asked to decide which of the two distributions was closer to the central fixation point (i.e., which had the statistical expectation closer to the center). They were asked to respond by shifting their gaze as quickly as possible onto the chosen, closer target. Observers were simply asked to 'look at the target': we did not explicitly require them to move to the mean of the generative distribution, nor to intercept the target's current location (we did not enforce an acceptance window; all saccades were included in the analysis as long as they left the fixation area and reduced the distance between gaze and one of the two distribution of target positions). Figure 1a (left sub-panel) illustrates one trial schematically: the observer is looking at the center of the screen (red trace), when the two targets appear and continue changing positions. Each trial started when gaze position was maintained within 2 deg from the central fixation point at least 200 ms. If the trial did not start within 2 s, the program paused, allowing participants to take a break and re-calibrate the eye-tracker. To prevent the use of monitor edges as stable landmarks for the localization of the peripheral targets, the position of the fixation point was jittered across trials: each trial a new position was drawn from a 2-D Gaussian distribution centered on the screen center, with a standard deviation of 0.2 deg on both horizontal and vertical dimension, and zero covariance. The position of the distributions from which the positions of the peripheral targets (the Gaussian blobs) were drawn was always clamped with respect to the trial-by-trial position of the fixation point. In any trial, the average distance of the centers of the two generative distributions was always 10 deg, but it differed across left and right targets, so that for one of the targets (the near target) it was always <10 deg, and for other >10 deg (see video S1 for an example). The difference in distance between the two distributions was adapted over the course of the experiment according to a two-down, one-up staircase procedure to achieve a similar level of performance (≈70% correct response) across participants. A 50-ms beep (F5, 698.46 Hz) was delivered as feedback after correct choices. Each participant ran a minimum of 20 blocks of 50 trials each, distributed over the course of several testing sessions on separate days. See Table S1 for information about the performances of individual observers.

*Experiment 2.* Experiment 2 followed a similar procedure to Experiment 1, but with the following differences. The generative distributions of target positions were both placed at the same distance: 10 deg of eccentricity. The perceptual decision was not based on the position of the targets, but on the average luminance of 2 squares, presented parafoveally, each containing 4 bars of varying luminance, re-sampled in synchrony with the peripheral target positions (see video S2 for an example). Participants were instructed to decide which of the two squares had higher average luminance and to communicate their decision in the same way as Experiment 1, that is by making a saccade to the target on the corresponding side of the screen. The luminance values of the bars were drawn from a Gaussian distribution (see "Stimuli" section), and the mean luminance of the brightest square was initialized at 8 cd/m² above the background luminance, and then adjusted according to a two-up one-down staircase procedure (step size 2 cd/m²). Each participant ran a minimum of 13 blocks of 50 trials each, distributed over the course of several testing sessions on separate days. Information about performances of individual observers is reported in Table S2.

*Experiment 3.* Experiment 3 followed a procedure similar to Experiment 1, with some differences in instructions, feedback, and distribution of target locations. As in Experiment 1, observers were required to identify the nearest target, and direct their gaze to it. However, in this case, they were explicitly instructed to direct their gaze to the mean of the distribution of the Gaussian blob locations (hereafter referred to as the 'centroid') on the chosen side. All naive participants in this experiment were experienced psychophysical observers, and before the beginning of the experiment they were briefed about the notions of 'mean' and 'centroid', and they were told explicitly that their task was to shift their gaze as close as possible to the centroid of the distribution of the nearest target's locations. In order to facilitate and encourage saccadic targeting of the centroid, we provided trial-by-trial feedback on the saccadic accuracy: after each trial, eye movement recordings were immediately analyzed to identify the endpoint of the primary saccade (defined as the first saccade that moved gaze by 2.5 deg or more away from the central fixation). We then displayed the estimated saccadic landing point alongside with the centroid location. In addition, the saccadic landing point was colored in green whenever the distance from the centroid was equal to or less than 1.25 deg, and red otherwise. Observers were asked to obtain as many 'greens' as possible (see Fig. S1b for examples of eye movements feedback). Finally, we also increased the external uncertainty about the centroid location, by adding some random trial-by-trial variations in the positions of the targets relative to fixation. Specifically, centroid positions were sampled along iso-eccentric semicircles (with small differences in the eccentricity of the left and right circles, adjusted by means of the two-down, one-up staircase procedure), with one position being anti-podal to the other along a line that passed through fixation and was tilted with a random angle uniformly distributed within ±30° from horizontal. Each participant ran a minimum of 20 blocks of 50 trials each, distributed over the course of several testing sessions on separate days. See Table S3 for information about the performances of individual observers.

## Analysis

*Pre-processing of gaze recordings.* Saccadic onsets and offsets were detected offline using MATLAB and an algorithm based on 2-D eye velocity[40]. More specifically,

eye movements were identified as saccades if their velocities exceeded the median velocity by 5 standard deviations for at least 8 ms. Once saccadic parameters were measured, further statistical analyses were made using the open-source software R (ref. [41]). For each trial, we selected as the primary saccade the first saccade that started after the onset of the target, from within a circular area of 2.5 deg around the initial fixation point, ended outside of that circular area. We excluded trials where the primary saccade had a latency shorter than 100 ms (≈0.5% of total trials in Experiment 1, ≈0.3% in Experiment 2, and ≈0.1% in Experiment 3) and trials where the amplitude of the primary saccade was less than 2.5 deg (≈5% of total trials in Experiment 1, ≈20% in Experiment 2, and ≈2% in Experiment 3).

*Estimation of weighting functions.* In order to estimate the weighting functions for saccade planning, we regressed the centers of gaze (with vertical and horizontal positions denoted $s_x$ and $s_y$) at saccadic termination against the spatio-temporal coordinates of the Gaussian blobs (temporally aligned with respect to the saccadic onset). We restricted our analysis to the 900 ms proceeding the onset of the eye movement. Since the granularity of saccadic onset detection was on the order of one millisecond, this yields 900 time points and thus, in principle, 900 parameters to estimate simultaneously. To make the estimation more tractable, we pooled the spatial coordinates into (100) 9-ms bins. Whenever changes in the position of the Gaussian blob occurred within a bin, we took the average of the two positions, weighted by the relative fraction of time in which the blob occupied each position within the bin. This procedure yields for each trial $i$ vectors of target positions $\mathbf{x}_i$ and $\mathbf{y}_i$, each of length 100. The trial-by-trial coordinates of saccadic endpoint were modeled as

$$s_{x,i} \sim \mathcal{N}(\alpha_x + \boldsymbol{\beta} \cdot \mathbf{x}_i, \sigma_x^2)$$

$$s_{y,i} \sim \mathcal{N}(\alpha_y + m\boldsymbol{\beta} \cdot \mathbf{y}_i, \sigma_y^2) \quad (1)$$

where $\boldsymbol{\beta} = (\beta_1, \beta_2, \ldots, \beta_{100})$ is the vector of linear coefficients determining which of the position samples are correlated with the saccadic landing position (assumed to be the same across vertical and horizontal saccadic components, up to a scaling factor $m$) and '·' is the dot product. Note that the linear coefficients are not independent from one another. Due to the temporal structure of the stimulus, contiguous coefficients often represent the influence of the same stimulus sample. This introduces autocorrelation in the coefficient vector, such that the difference between neighboring coefficients is likely to be smaller than that of more distant coefficients. To account for this, we fit our model within a Bayesian framework and adopted a random-walk prior[42] to enforce smoothness:

$$\beta_{100} \sim \mathcal{N}(0, 0.1)$$

$$\beta_i \sim \mathcal{N}(\beta_{i+1}, \tau)$$

$$\tau \sim \mathcal{N}(0, 0.1) \quad (2)$$

Note that the random-walk proceeds in reverse—starting by assigning a regularizing (zero-centered) Gaussian prior to the last coefficient. This is because the last coefficient lies within 9 ms from the saccade onset, and thus is unlikely to have a large influence on the saccadic vector. The remaining parameters were assigned the following priors

$$\alpha_x, \alpha_y \sim \mathcal{N}(0, 1)$$

$$\sigma_x, \sigma_y \sim \text{HalfCauchy}(0, 1)$$

$$m \sim \mathcal{N}(1, 1). \quad (3)$$

This modeling approach was used in all 3 experiments. For each participant, the model was estimated using MCMC sampling in Stan and its R interface[43]. We ran 4 chains of 4000 samples each and verified convergence by checking that there were no divergent transitions and the variance between and within chains did not differ significantly; $\hat{R} \approx 1$ for all parameters[44].

A similar approach was used to estimate the weighting function for the decision, with the difference that we used a generalized linear model instead of a simple linear regression, to account for the dichotomous nature of the dependent variable. Formally, in this model, the probability of choosing the stimulus on the right can be expressed as

$$P(\text{choose right}) = \Phi\left(\frac{\alpha + \boldsymbol{\beta} \cdot \Delta}{\sqrt{2}\sigma}\right) \quad (4)$$

where, for Experiment 1 and Experiment 3,

$$\Delta = \left[\mathbf{x}_{\text{right}}^{\circ 2} + \mathbf{y}_{\text{right}}^{\circ 2}\right]^{\circ\frac{1}{2}} - \left[\mathbf{x}_{\text{left}}^{\circ 2} + \mathbf{y}_{\text{left}}^{\circ 2}\right]^{\circ\frac{1}{2}} \quad (5)$$

is the vector of differences between the two targets' distances from the central fixation point. This vector contains 100 values for each trial (for clarity we omitted the trial subscript $i$). The notation '∘' in the exponents indicates that the power operations are applied elementwise (also known as Hadamard power). The same approach was used in the analysis of Experiment 2, however, in this case the perceptual decision was based on the difference in luminance between the right and

left patch,

$$\mathbf{\Delta} = \mathbf{L}_{\text{right}} - \mathbf{L}_{\text{left}} \qquad (6)$$

where $\mathbf{L}$ indicates the vector of luminances of either the left or right parafoveal patch (each value represents the average of the 4 vertical bars within the patch). Before estimating the model, luminance differences were re-scaled to have approximately the same standard deviation as position differences in Experiment 1 (this enabled us to plot the resulting weighting function onto a similar scale, Fig. 1).

To introduce smoothness we used, for both experiments, the same random-walk prior used in the analysis of saccadic weighting functions. The remaining parameters were given the following priors

$$\alpha \sim \mathcal{N}(0, 1)$$

$$\sigma \sim \text{HalfCauchy}(0, 1). \qquad (7)$$

This model was also estimated using MCMC sampling as implemented in Stan.

*Statistical tests.* The ordered vector of 100 coefficients represents an estimate of the weighting function used by participants to make the decision and to plan the eye movement. In order to estimate the onsets and offsets of the temporal integration windows, for each participant, we used samples drawn from the posterior distribution to estimate the Bayesian highest posterior density (HPDI) credible intervals around each of the 100 coefficients. This allowed us to determine the temporal integration windows as the temporal intervals in which the credible interval did not include zero. To control further for the possibility that these intervals were due to chance, we estimated their probability under the null hypothesis using the cluster test[45,46]. For this test, each coefficient was transformed into a $t$ statistic by dividing it by the standard deviation of its posterior distribution. The number of resolution elements or resels (which determines the resolution of the random field assumed by the cluster test) was taken to be the number of distinct stimulus samples presented during the 900 ms interval before the saccade: 13.5. For all the clusters included in the analysis, the $p$-value resulting from this procedure was smaller than 0.01. To determine onsets and offsets of the integration windows at the group level, we averaged the onset and offset of the integration windows of individual participants (see Fig. 1).

**Reporting summary**. Further information on research design is available in the Nature Research Reporting Summary linked to this article.

## Data availability

Data and code supporting this article are available as an Open Science Framework repository (link: https://osf.io/embky/)[47]. Additional data underlying the main and Supplementary Figures is available in Supplementary Data 1–5.

## Code availability

Data and code supporting this article are available as an Open Science Framework repository (link: https://osf.io/embky/)[47].

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

## Acknowledgements

This work was supported by grant RPG-2016-124 from the Leverhulme Trust.

## Author contributions

All three authors developed the concept and contributed to the study design. M.L. collected and analyzed the data. M.L. wrote the manuscript and J.A.S. and M.J.M. edited it. All authors gave final approval for publication.

## Competing interests

The authors declare no competing interests.
