## [Peer Review File · Communications Biology]

Reviewers' comments:

Reviewer #1 (Remarks to the Author):

I generally liked this paper. I think there are important issues of interpretation (see below), but overall I found the experiments to be well-designed, and the question is interesting. It is also a difficult question to answer conclusively, so I appreciated that the authors took it on and did their best at it. The analysis is a bit circumvolved and may appear unnecessarily complicated at times, and sure, one could start poking at it from various angles, but I don't think it would be useful because it is in many ways sensible, and I do believe the data: it looks like the effect is real, at least the effect exposed by the analysis. My problem lies with the interpretation.

First, there are some hidden details which seem to complicate interpretation, and these are overlooked by the way data is plotted in Figure 1. Take panel e: the quantity plotted on the y axis is directly comparable for black and blue traces, despite the fact that the black trace involves a logistic conversion-to-response: at least, for both traces, 'weight' is regressed against the position of the Gaussian blob. This is not the case, however, for panel f, or at least that is not what I understood from reading Methods: in panel f, 'weight' on the y axis is regressed against the position of the Gaussian blob for the blue trace, while it is regressed against the brightness of the parafoveal patch for the black trace. Now, that is somewhat problematic for interpretation (there is also another puzzling feature which I mention further below). I like the fact that it is consistent with Figure 1e, but at the same time I think readers should be alerted to this distinction (unless I completely misunderstood what is being plotted). And this brings me to the main interpretational issue I have with this paper, which is in part related to the above observation. The authors discuss it at lines 345-348: it is possible that the design of both experiments forced (or at least pushed) observers to program eye movement only after target selection. Basically the issue here is that decision and saccade are two distinct problems/tasks as defined by design: decision is a centroid judgment, saccade is an instantaneous position estimate. The former is, by design, meant to involve data collection over an extended window of time. The latter is, again by design, meant to involve data collection only over a short time before taking action. If one were to take an unfriendly view, one would conclude that the outcome of these experiments is pretty much dictated by design choice and therefore trivial, and tells us little about the potential temporal dissociation between decision and action. The fact that Exp 2 aligns with Exp 1 actually goes to support this view: in Exp 2, the separation-by-design between decision and motor action is even clearer than in Exp 1. The results are the same, suggesting that the correct interpretation of Exp 1 is to think of it as a more subtle version of Exp 2.

To clarify the above, let's consider a design that would retain a closer relationship between decision and saccade: in this version of Exp 1, observers are asked to make a saccade to the estimated centroid of the blob distribution, i.e. to the same spatial quantity upon which the decision is based. I accept that this is a tall order for an observer: the experimenter is asking the observer to make a saccade to an unseen target. To make things worse, this would need to happen in the presence of an actual target nearby, which certainly complicates interpretation. One could get around this last point by removing the Gaussian blob as soon as the saccade is initiated, so that saccadic landing would not find a Gaussian blob. But the observer is still saccading to an estimated location without landing target, so it is unclear whether a protocol of this kind would work: it may just be too noisy/difficult to extract useful information. And I am sure the authors have thought of this variant, and have found a good reason why the experiment could not be done that way. Still, the point of considering this version of the design is to clarify the interpretational issues associated with the paradigm that was used in the study. Had the experiment been designed this way, it would make a more compelling case, at least for me.

As I said, the authors did consider the above issue in Discussion, however they do not really address it. The relevant paragraph is written in an odd way: first, the authors bring up the issue (lines 345-348). Instead of addressing it, they bring up another issue at lines 348-351 relating to fovea versus periphery. Over the following lines 351-360, they proceed to address the latter issue, but not the former. By the end of this paragraph, readers are left with no answer. I said this paragraph reads oddly because when I first read it I somehow had the impression that the authors had addressed the issue, but I couldn't explain to myself how, then I went back, re-read it, and

realized that in fact they had not explained THAT issue, only the other one. In other words, the way to write this would be to introduce a paragraph break at line 348, so that a new paragraph is started with "Moreover". The paragraph that remains above it, then, obviously leaves the most important issue unaddressed.

The above is really the main sticking point for me. I understand the general question is difficult to address experimentally, and the authors should be commended for having taken a good shot at it, but unless they can address the above issue satisfactorily, I remain unconvinced.

Puzzling features:

Figure 1f: the black trace goes negative for x values below -0.65. How do we interpret this? It is not clear what a negative weight means, except perhaps that it is some sort of differential operator for luminance, like a temporal contrast detector, but those objects operate on a much faster scale in vision. Perhaps worth discussing.

Figure 2: I am not sure I fully understand how data for this Figure was computed, because I would expect that there should be no data to the left of about -0.7 on the x axis in the left-bottom panel: doesn't this panel refer to trials on which saccade onset is distributed as in the panel immediately above it? If so, there should be no jumping Gaussian blobs to the left of that distribution, which leaves me wondering as to how the weight was computed there.

Minor:

line 73 states that 6 participants took part in Exp 2, but Figure 1f only plots 5 significance bars at the bottom. I understand from the caption to Figure 2 that some participants do not show up as bars when they do not reach significance, but maybe specify this in the caption to Figure 1 too, otherwise readers will be puzzled by this potential discrepancy.

line 138, when describing Exp 2: it is stated that the perceptual decision was based on a different stimulus, but readers are never told what that decision was supposed to be. They will find it in the caption to Figure 1, where it states that they were asked to choose the brightest patch. But this specification should be available from Methods. Also, when reading that section in Methods, I was not entirely clear as to whether observers were asked to move their eyes to the patch, or to the Gaussian blob like in Exp 1. It does become clearer that the latter applies when I read the caption to Figure 1 but, once again, this should be absolutely clear from reading the Methods section.

line 276: Figure 3 is described as an example of speed-accuracy tradeoff. In a sense, it is. But it goes in the opposite direction of what is normally found: the phenomenon commonly termed speed-accuracy tradeoff (which also takes the acronym SAT) involves a positive relationship between reaction time and accuracy, not negative like in Figure 3. This is not a problem: it's just data, and if that's what the data look like, fine. It's also ok to call it speed-accuracy tradeoff: it is a tradeoff of some sort, even if it goes in the opposite direction. However I think it would help readers if this aspect were discussed, so that they may not be puzzled when they look at Figure 3. Some readers may not even bother looking at Figure 3 under the assumption that they know what SAT is. They would then gather incorrect information from the text.

Figure 1e-f: the label 'perception' is slightly unsettling when referring to the black trace. I understand what the authors mean and are trying to achieve with that, but I am not entirely happy with it. If you guys want to go all the way in that direction, then replace the label 'saccade' with 'action'. If you want to keep saccade, though, then I think the other label needs to be 'decision'. You either keep it sober, or you go all the way sexy.

Reviewer #2 (Remarks to the Author):

In this paper the authors examine whether decision formation and motor preparation during speeded perceptual decisions are distinct or overlapping. In two experiments, involving the serial accumulation of visual information and the communication of the decision via a saccade, the

authors report that decision-formation and motor preparation are influenced by early and late information, respectively. Interestingly, the temporal weighting functions corresponding to the categorical decision and the saccadic endpoint, do not overlap temporally. The authors interpret this finding as supporting the idea that decision-formation and motor preparation are distinct processes.

Overall, the question addressed in the paper is timely and interesting. As concisely described in the Introduction, the idea that there is a common neural representation leading to a decision and to the subsequent motor execution found support in early influential neurophysiological experiments. However, this very idea has been more recently questioned. The current paper aims to contribute to this debate by employing psychophysical experiments and rigorous statistical analyses.

I do not have substantial comments on the technical aspects of the manuscript. The experiments and subsequent analyses are carefully designed and executed.

My main concern is rather conceptual. Specifically, I am not entirely convinced that the two paradigms employed here are suitable to test whether decision-formation and motor-preparation are independent or not. The reason for my assessment is that the location of the saccadic target is not stable within a trial. Even though participants are not encouraged to "intercept the target" and even though any saccade to the side of the target is acceptable, the most recent location of the Gaussian blob can indeed act as the saccadic target. This would be consistent with the rather transient saccadic weighting function and the lack of overlap between the perception and saccadic functions (in the sense that the saccadic target is not precisely known during the decision formation). Please note that I understand that nominally, the target was known throughout the trial and that it corresponds to any location on the relevant side of the fixation. I just think that participants involuntarily anchored their saccade to the more recent blob location and treat that as the target.

Experiment 2 more clearly reveals the point I am raising here. The categorical response is based on different sensory stimuli while the targets (left and right) fluctuate in time and space. From the viewpoint of the participant, accumulating the target location over time is not relevant. Nevertheless, the individual moving targets could act as irrelevant anchors, towards which participants saccade after they formed a decision. A more suitable paradigm would be very similar to Experiments 1/2 but would require participants to saccade onto the average target location (and receive feedback both based on the categorical correctness and on how close they hit the average point). That way, the two weighting functions could be more easily interpretable and indeed capable of revealing whether decision-formation and motor-planning evolve sequentially or in parallel.

All in all, I believe that with the current experiments strong conclusions cannot be drawn and that the reported dissociation between decision formation and motor planning is a byproduct of the experimental design.

Reviewer #3 (Remarks to the Author):

This manuscript presents psychophysical data from two (similar) experiments in which participants view two streams of dynamic stimuli, one on the left and another on the right, compare them in accordance to task instructions, and make an eye movement to the side that corresponds to the correct decision. Then, analyses akin to reverse correlation are made to determine how strongly the choices were influenced by the stimuli that preceded them at each point in time, from early in the trial (up to ~700 ms) to late in the trial (just before saccade onset). The main result is that

"...the temporal weighting functions for decision-formation and motor-preparation were distinct and largely non-overlapping: whereas early samples predicted the perceptual decision, later samples predicted only the parameters of the saccade but not the decision."

This would indeed be a very provocative finding because, as pointed out in the text, it would go

against the standard framework in which perceptual decisions and motor responses share similar neural substrates (at least for saccadic choices). In fact, I have strong reason to think that oculomotor structures are **not** the sites where perceptual evaluations take place. Unfortunately, however, the conclusion is based on wrong assumptions and does not follow from the data. The claimed dissociation between perceptual and motor processes never actually happened.

In the experiments there is only one choice per trial, one decision that is indicated by a saccade. Then different weighting functions are computed for saccade direction and saccade landing position. These turn out to be different. Then 'saccade direction' is equated with 'perceptual decision' and 'saccade landing position' is equated with 'motor preparation' --- so therefore, "decision formation" and "oculomotor responses" have been dissociated.

The crucial, wrong assumption is that the specification of saccade direction many hundreds of milliseconds before the saccade onset is not part of a motor planning process, or that if there is motor planning activity specifying the saccade direction at that point, such activity must also determine the saccade landing point. Decades of oculomotor neurophysiology are inconsistent with either of these scenarios.

Consider a delayed saccade task: the participant fixates on a central spot; then a saccade target is shown, but the participant must keep fixating for 1 s; then the fixation point disappears, instructing the participant to look at the target. Here, the saccade direction is specified 1 s before the actual movement, and the target and trigger stimuli are different. But is there a motor plan during that delay interval? Yes, absolutely. During such delays (whether for visually- or memory-guided tasks), oculomotor neurons of the visual (V) and visuomotor (VM) types in FEF, LIP, and SC fire in a sustained way, clearly encoding the direction of the future movement. Does this activity determine the exact saccade landing position? Probably not, or just very weakly. After the go signal (fixation point offset), the neurons classified as motor (M) become strongly active, and the rise in their activity determines the onset of the saccade and the observed saccade metrics (peak speed, amplitude, etc.). The delay activity, which is clearly indicative of saccade direction, will account for little of the variance in the landing positions.

So, oculomotor plans specifying a future saccade vector can easily develop several seconds before the activity that determines the onset of the upcoming saccade and its specific kinematics. This means that the results do not dissociate "decision formation" and "oculomotor responses"; they simply dissociate saccade direction from saccade metrics. This also makes perfect sense because in this task the saccade 'target' is constantly changing. But regardless, this is all completely inconsequential to the intentional framework, because it does not distinguish motor planning from decision formation.

Reviewers' comments.
(Our responses in bold font)

Reviewer #1 (Remarks to the Author):

I generally liked this paper. I think there are important issues of interpretation (see below), but overall I found the experiments to be well-designed, and the question is interesting. It is also a difficult question to answer conclusively, so I appreciated that the authors took it on and did their best at it. The analysis is a bit circumvolved and may appear unnecessarily complicated at times, and sure, one could start poking at it from various angles, but I don't think it would be useful because it is in many ways sensible, and I do believe the data: it looks like the effect is real, at least the effect exposed by the analysis. My problem lies with the interpretation.

First, there are some hidden details which seem to complicate interpretation, and these are overlooked by the way data is plotted in Figure 1. Take panel e: the quantity plotted on the y axis is directly comparable for black and blue traces, despite the fact that the black trace involves a logistic conversion-to-response: at least, for both traces, 'weight' is regressed against the position of the Gaussian blob. This is not the case, however, for panel f, or at least that is not what I understood from reading Methods: in panel f, 'weight' on the y axis is regressed against the position of the Gaussian blob for the blue trace, while it is regressed against the brightness of the parafoveal patch for the black trace. Now, that is somewhat problematic for interpretation (there is also another puzzling feature which I mention further below). I like the fact that it is consistent with Figure 1e, but at the same time I think readers should be alerted to this distinction (unless I completely misunderstood what is being plotted).

We thank the reviewer for pointing to this possible source of confusion. Firstly, we should stress that the absolute heights of the black and blue traces in panel 1e are not directly comparable, precisely for the reasons mentioned by the reviewer (the black trace involves a non-linear transformation from the unbounded space of the model coefficients to the bounded probability measure of the binary response, therefore they are not expressed in the same units). However, they do happen to have similar magnitudes, which enabled us to plot them alongside each other without having to do any rescaling (this point is now clarified mentioned in the figure caption, pag 14 lines 16-17).

The black trace in panel 1f gives the weighting of parafoveal patch brightnesses (specifically the luminance differences between left and right) and, as the reviewer correctly pointed out, it should not be expected to have a magnitude similar to the trace in 1e. The reason for similar magnitudes is that, prior to further analyses, the luminance differences in Exp. 2 were rescaled to have approximately the same standard deviation as the position differences in Exp. 1. This initial step of the analysis, while being documented in the Github repository containing code and data, was omitted from the Methods section, so we thank the reviewer for noting this small inconsistency. This detail has been included in the revised version of the Methods section (pag. 10 lines 15-17).

And this brings me to the main interpretational issue I have with this paper, which is in part related to the above observation. The authors discuss it at lines 345-348: it is possible that the design of both experiments forced (or at least pushed) observers to program eye movement only after target selection. Basically the issue here is that decision and saccade are two distinct problems/tasks as defined by design: decision is a centroid judgment, saccade is an instantaneous position estimate. The former is, by design, meant to involve data collection over an extended window of time. The latter is, again by design, meant to involve data collection only over a short time before taking action. If one were to take an unfriendly view, one would conclude that the outcome of these

experiments is pretty much dictated by design choice and therefore trivial, and tells us little about the potential temporal dissociation between decision and action. The fact that Exp 2 aligns with Exp 1 actually goes to support this view: in Exp 2, the separation-by-design between decision and motor action is even clearer than in Exp 1. The results are the same, suggesting that the correct interpretation of Exp 1 is to think of it as a more subtle version of Exp 2.

To clarify the above, let's consider a design that would retain a closer relationship between decision and saccade: in this version of Exp 1, observers are asked to make a saccade to the estimated centroid of the blob distribution, i.e. to the same spatial quantity upon which the decision is based. I accept that this is a tall order for an observer: the experimenter is asking the observer to make a saccade to an unseen target. To make things worse, this would need to happen in the presence of an actual target nearby, which certainly complicates interpretation. One could get around this last point by removing the Gaussian blob as soon as the saccade is initiated, so that saccadic landing would not find a Gaussian blob. But the observer is still saccading to an estimated location without landing target, so it is unclear whether a protocol of this kind would work: it may just be too noisy/difficult to extract useful information. And I am sure the authors have thought of this variant, and have found a good reason why the experiment could not be done that way. Still, the point of considering this version of the design is to clarify the interpretational issues associated with the paradigm that was used in the study. Had the experiment been designed this way, it would make a more compelling case, at least for me.

As I said, the authors did consider the above issue in Discussion, however they do not really address it. The relevant paragraph is written in an odd way: first, the authors bring up the issue (lines 345-348). Instead of addressing it, they bring up another issue at lines 348-351 relating to fovea versus periphery. Over the following lines 351-360, they proceed to address the latter issue, but not the former. By the end of this paragraph, readers are left with no answer. I said this paragraph reads oddly because when I first read it I somehow had the impression that the authors had addressed the issue, but I couldn't explain to myself how, then I went back, re-read it, and realized that in fact they had not explained THAT issue, only the other one. In other words, the way to write this would be to introduce a paragraph break at line 348, so that a new paragraph is started with "Moreover". The paragraph that remains above it, then, obviously leaves the most important issue unaddressed.

The above is really the main sticking point for me. I understand the general question is difficult to address experimentally, and the authors should be commended for having taken a good shot at it, but unless they can address the above issue satisfactorily, I remain unconvinced.

We thank the reviewer for this observation. We agree that there two issues at stake, which relate to two alternative explanations for the serial pattern seen in the weighting functions that we measured. The first possibility, which is the one we intended to address, is that peripheral processing stops shortly before a saccade is launched (as suggested by Ludwig et al., 2014). We believe that this possibility is fully addressed by our Experiment 2, in which the perceptual decision is based on visual information presented parafoveally. However, the previous version of the manuscript did not fully address the other issue, which is the main concern of the reviewer. The reviewer correctly noted that integrating spatial positions over time toward the calculation of a centroid is, strictly speaking, not required in the saccadic task – since we only asked them to “saccade to the target,” observers might as well saccade to the last position registered before the saccadic dead time. The reviewer observed that the temporal separation of evidence assimilation for action and perceptual decision that we have observed could have been a consequence of the task instructions, and therefore our results would leave open the possibility that saccadic programming may change if observers were instead required to saccade to the centroid.

This latter explanation cannot fully account for the pattern seen in our data. If saccadic programming were simultaneous with the integration of perceptual evidence, then we should expect some degree of overlap between saccadic and perceptual weighting functions, even if observers merely aimed their saccades toward a single target position. Nevertheless, we concede that it might be difficult to predict the extent of that overlap precisely. Thus, in order to rule out this explanation conclusively, we have now explicitly tested the condition in which observers are instructed to saccade at the centroid (the mean of the blob's distribution). This is our new Experiment 3. (Note that due to the Covid-19 pandemic we were able to collect the data only recently; after restrictions were lifted.) As reported in the text, the results of Experiment 3 fully replicate the pattern seen in Experiments 1 and 2, thus corroborating our original interpretation.

In Experiment 3 observers were all experienced psychophysical observers, familiar with the notion of 'centroid', and were explicitly required to saccade to the centroid. To encourage this, we provided a trial-by-trial feedback on their saccadic accuracy: after each response, eye movements were analyzed online and the estimated saccadic landing point was displayed alongside with the true centroid of the blob's distribution. The saccadic landing point was coloured in green whenever the distance from the centroid was less than 1.25 deg, and red otherwise; observers were asked to obtain as many 'greens' as possible (see below for an example). Another small change from the design of Experiment 1 is that, in order to avoid participants being able to saccade to two 'default' locations to the right and left to the fixation point, we added some random vertical variation: centroid positions were sampled along iso-eccentric semicircles (with small differences in the eccentricity of the left and right circles, adjusted by means of the staircase procedure), with one position being antipodal to the other along a line that passed through fixation and was tilted with a random angle within $\pm 30^\circ$ from horizontal. We note that despite the additional positional uncertainty induced by this manipulation, observers achieved a similar or even slightly smaller average saccadic errors than in Experiment 1: the mean error in Experiment 3 was 2.07 deg (SD 0.34), while it was 2.44 (SD 0.76) in Experiment 1. Furthermore, comparing the variance of saccadic landings to previous data that used a similar (but static) visual stimulus suggested that observers used more than one position sample to direct their saccades (see Fig. S2).

Eye movement feedback in Experiment 3. In both the left and the right panels, the participant made the correct perceptual decision, as indicated by the fact that saccadic endpoints (large green and red dots) and the revealed centroids (white dots) are on the same side of fixation (small black box). In the left panel, the saccade endpoint is colored in green as it was less than 1.25 deg away from the centroid, whereas on the right panel it is colored in red to indicate that the subject's saccade should have been more accurate. In both panels the smaller, darker green and red dots show all gaze samples from target onset to saccade completion.

The reviewer wondered whether the presence of a visible target after the saccade was launched would be an issue for the new Experiment. We don't think this is a concern or a confound, since saccadic programming is largely ballistic in nature and the saccadic motor program cannot in general be modified after it is launched. Moreover, the presence of a visible target *after* saccadic landing is also not a concern in our task: given the statistical structure of

targets in our experiment, the centroid is the location that minimizes the expected post-saccadic error between gaze position and target.

Puzzling features:

Figure 1f: the black trace goes negative for x values below -0.65. How do we interpret this? It is not clear what a negative weight means, except perhaps that it is some sort of differential operator for luminance, like a temporal contrast detector, but those objects operate on a much faster scale in vision. Perhaps worth discussing.

Well spotted. A negative weight would mean that a lower-than-average luminance presented early in the stimulus sequence would increase (rather than decrease) the probability of a patch being selected as the brightest. This could be due to a hysteresis effect, which is known to affect brightness judgments (e.g. see Stevens, 1961), in which the early dark (or light) patches would make the subsequent ones appear lighter (or darker).

Stevens, S. S. (1961). To honor Fechner and repeal his law. *Science*. 133(3446):80–86.

Figure 2: I am not sure I fully understand how data for this Figure was computed, because I would expect that there should be no data to the left of about -0.7 on the x axis in the left-bottom panel: doesn't this panel refer to trials on which saccade onset is distributed as in the panel immediately above it? If so, there should be no jumping Gaussian blobs to the left of that distribution, which leaves me wondering as to how the weight was computed there.

Indeed, the data provide little or no information for constraining the weights in that region (to the left of ~ -0.7 sec). The estimated value of the weights in that region is determined by the random-walk prior, whose mean is given - for each timepoint - by the estimated weight at the immediately following time point (which is why the line remains at a constant height there (to the left of ~ -0.7 sec). To avoid confusion, in the updated version of the manuscript, we plotted weights only at time points for which data were available in at least 5% of the trials for each observer.

Minor:

line 73 states that 6 participants took part in Exp 2, but Figure 1f only plots 5 significance bars at the bottom. I understand from the caption to Figure 2 that some participants do not show up as bars when they do not reach significance, but maybe specify this in the caption to Figure 1 too, otherwise readers will be puzzled by this potential discrepancy.

Thank you for spotting this – there was a typo in the manuscript at line 73: only 5 observers participated in Exp. 2 (1 author and 4 naive observers). This has been fixed in the updated version of the manuscript.

line 138, when describing Exp 2: it is stated that the perceptual decision was based on a different stimulus, but readers are never told what that decision was supposed to be. They will find it in the caption to Figure 1, where it states that they were asked to choose the brightest patch. But this specification should be available from Methods. Also, when reading that section in Methods, I was not entirely clear as to whether observers were asked to move their eyes to the patch, or to the Gaussian blob like in Exp 1. It does become clearer that the latter applies when I read the caption to Figure 1 but, once again, this should be absolutely clear from reading the Methods section.

We now provide more details about the instructions and the procedure within the Method section (page 6, starting from line 26). The description of the procedure of Experiment 2 now reads: “Experiment 2 followed a similar procedure to Experiment 1, but with the following differences. The generative distributions of target positions were both placed at the same distance: 10 deg of eccentricity. The perceptual decision was not based on the position of the targets, but on the average luminance of 2 squares, presented parafoveally, each containing 4 bars of varying luminance, resampled in synchrony with the peripheral target positions (see video S2 for an example). Participants were instructed to decide which of the two squares had higher average luminance and to communicate their decision in the same way as Experiment 1, that is by making a saccade to the target on the corresponding side of the screen.”

line 276: Figure 3 is described as an example of speed-accuracy tradeoff. In a sense, it is. But it goes in the opposite direction of what is normally found: the phenomenon commonly termed speed-accuracy tradeoff (which also takes the acronym SAT) involves a positive relationship between reaction time and accuracy, not negative like in Figure 3. This is not a problem: it's just data, and if that's what the data look like, fine. It's also ok to call it speed-accuracy tradeoff: it is a tradeoff of some sort, even if it goes in the opposite direction. However I think it would help readers if this aspect were discussed, so that they may not be puzzled when they look at Figure 3. Some readers may not even bother looking at Figure 3 under the assumption that they know what SAT is. They would then gather incorrect information from the text.

Thanks for pointing this out. We agree that our data do not contain evidence of a traditional speed-accuracy trade-off or SAT. We now describe this pattern of results as a relationship between speed and accuracy (page 13, lines 7-9). We suggest that this pattern of results is likely due to trial-by-trial fluctuations in the decision difficulty (due to the staircase procedure).

Figure 1e-f: the label 'perception' is slightly unsettling when referring to the black trace. I understand what the authors mean and are trying to achieve with that, but I am not entirely happy with it. If you guys want to go all the way in that direction, then replace the label 'saccade' with 'action'. If you want to keep saccade, though, then I think the other label needs to be 'decision'. You either keep it sober, or you go all the way sexy.

Agreed. We have relabelled the black line as 'decision' as it seems more appropriate in this context.

Reviewer #2 (Remarks to the Author):

In this paper the authors examine whether decision formation and motor preparation during speeded perceptual decisions are distinct or overlapping. In two experiments, involving the serial accumulation of visual information and the communication of the decision via a saccade, the authors report that decision-formation and motor preparation are influenced by early and late information, respectively. Interestingly, the temporal weighting functions corresponding to the categorical decision and the saccadic endpoint, do not overlap temporally. The authors interpret this finding as supporting the idea that decision-formation and motor preparation are distinct processes.

Overall, the question addressed in the paper is timely and interesting. As concisely described in the Introduction, the idea that there is a common neural representation leading to a decision and to the subsequent motor execution found support in early influential neurophysiological experiments. However, this very idea has been more recently questioned. The current paper aims to contribute to this debate by employing psychophysical experiments and rigorous statistical analyses.

I do not have substantial comments on the technical aspects of the manuscript. The experiments and subsequent analyses are carefully designed and executed.

My main concern is rather conceptual. Specifically, I am not entirely convinced that the two paradigms employed here are suitable to test whether decision-formation and motor-preparation are independent or not. The reason for my assessment is that the location of the saccadic target is not stable within a trial. Even though participants are not encouraged to “intercept the target” and even though any saccade to the side of the target is acceptable, the most recent location of the Gaussian blob can indeed act as the saccadic target. This would be consistent with the rather transient saccadic weighting function and the lack of overlap between the perception and saccadic functions (in the sense that the saccadic target is not precisely known during the decision formation). Please note that I understand that nominally, the target was known throughout the trial and that it corresponds to any location on the relevant side of the fixation. I just think that participants involuntarily anchored their saccade to the more recent blob location and treat that as the target.

Experiment 2 more clearly reveals the point I am raising here. The categorical response is based on different sensory stimuli while the targets (left and right) fluctuate in time and space. From the viewpoint of the participant, accumulating the target location over time is not relevant. Nevertheless, the individual moving targets could act as irrelevant anchors, towards which participants saccade after they formed a decision. A more suitable paradigm would be very similar to Experiments 1/2 but would require participants to saccade onto the average target location (and receive feedback both based on the categorical correctness and on how close they hit the average point). That way, the two weighting functions could be more easily interpretable and indeed capable of revealing whether decision-formation and motor-planning evolve sequentially or in parallel.

All in all, I believe that with the current experiments strong conclusions cannot be drawn and that the reported dissociation between decision formation and motor planning is a byproduct of the experimental design.

We thank the reviewer for their comments, which resonate with the concern voiced by Reviewer 1. As we explain above in response to reviewer 1, participants simply could have made saccades toward the most recent location of the Gaussian blob target. While we believe this interpretation cannot fully explain the lack of overlap between the measured weighting functions (see our reply to R1 comment), we have run a new Experiment 3 with feedback on both the categorical correctness and the saccadic error relative to the centroid (the average target position). Despite the different instructions and the trial-by-trial feedback on saccadic

accuracy, we fully replicate the pattern of non-overlap between weighting functions, thus corroborating our interpretation of a temporal dissociation between decision formation and saccade programming. Please see the reply to R1 and the main text for more details on Experiment 3.

Reviewer #3 (Remarks to the Author):

This manuscript presents psychophysical data from two (similar) experiments in which participants view two streams of dynamic stimuli, one on the left and another on the right, compare them in accordance to task instructions, and make an eye movement to the side that corresponds to the correct decision. Then, analyses akin to reverse correlation are made to determine how strongly the choices were influenced by the stimuli that preceded them at each point in time, from early in the trial (up to ~700 ms) to late in the trial (just before saccade onset). The main result is that

"...the temporal weighting functions for decision-formation and motor-preparation were distinct and largely non-overlapping: whereas early samples predicted the perceptual decision, later samples predicted only the parameters of the saccade but not the decision."

This would indeed be a very provocative finding because, as pointed out in the text, it would go against the standard framework in which perceptual decisions and motor responses share similar neural substrates (at least for saccadic choices). In fact, I have strong reason to think that oculomotor structures are **not** the sites where perceptual evaluations take place. Unfortunately, however, the conclusion is based on wrong assumptions and does not follow from the data. The claimed dissociation between perceptual and motor processes never actually happened.

In the experiments there is only one choice per trial, one decision that is indicated by a saccade. Then different weighting functions are computed for saccade direction and saccade landing position. These turn out to be different. Then 'saccade direction' is equated with 'perceptual decision' and 'saccade landing position' is equated with 'motor preparation' --- so therefore, "decision formation" and "oculomotor responses" have been dissociated.

The crucial, wrong assumption is that the specification of saccade direction many hundreds of milliseconds before the saccade onset is not part of a motor planning process, or that if there is motor planning activity specifying the saccade direction at that point, such activity must also determine the saccade landing point. Decades of oculomotor neurophysiology are inconsistent with either of these scenarios.

Consider a delayed saccade task: the participant fixates on a central spot; then a saccade target is shown, but the participant must keep fixating for 1 s; then the fixation point disappears, instructing the participant to look at the target. Here, the saccade direction is specified 1 s before the actual movement, and the target and trigger stimuli are different. But is there a motor plan during that delay interval? Yes, absolutely. During such delays (whether for visually- or memory-guided tasks), oculomotor neurons of the visual (V) and visuomotor (VM) types in FEF, LIP, and SC fire in a sustained way, clearly encoding the direction of the future movement. Does this activity determine the exact saccade landing position? Probably not, or just very weakly. After the go signal (fixation point offset), the neurons classified as motor (M) become strongly active, and the rise in their activity determines the onset of the saccade and the observed saccade metrics (peak speed, amplitude, etc.). The delay activity, which is clearly indicative of saccade direction, will account for little of the variance in the landing positions.

So, oculomotor plans specifying a future saccade vector can easily develop several seconds before the activity that determines the onset of the upcoming saccade and its specific kinematics. This means that the results do not dissociate "decision formation" and "oculomotor responses"; they simply dissociate saccade direction from saccade metrics. This also makes perfect sense because in this task the saccade 'target' is constantly changing. But regardless, this is all completely inconsequential to the intentional framework, because it does not distinguish motor planning from decision formation.

We thank the reviewer for their thoughtful evaluation of our manuscript, which helped us to identify aspects that required clarification and further elaboration.

The reviewer noted that the early processing phase, which we refer to as perceptual decision formation, can be seen as an early, coarse phase of motor programming - determining whether the saccade will be directed toward the left or right hemifield. Therefore, the temporal dissociation between perceptual decision and oculomotor responses might not be as clearly defined as we suggest.

We agree with the reviewer that early processing could be understood as coarse motor planning – this was also acknowledged in the previous version of the manuscript (Discussion, lines 356-58: “...we cannot exclude the possibility that relatively early samples (>200 msec before the saccade) may have suggested spatially imprecise motor plans...”). Nonetheless, our conclusions remain valid. Rather than assuming early processing is non-motor, our results demonstrate the presence of late, “non-decisional” processing (in the sense that it occurs after the observer committed to a perceptual decision). This is demonstrated by the finding that the precise coordinates of saccadic endpoints were determined by later samples of stimulus; samples that did not influence the dichotomous left/right decision. Although our results may not completely contradict the idea that “*perceptual decision making is implemented in the brain as a process of choosing between available motor actions*” (Shushruth, Mazurek & Shadlen, 2018, J Neurosci), they do unambiguously imply (i) that motor actions are not yet ready to launch when the observer commits to a decision in speeded perceptual decision tasks and (ii) that visual input received after decision commitment can still feed into the motor system and influence the saccadic response. While it is known that post-decisional processing in humans can influence metacognitive processes such as the confidence in the response (e.g., see Navajas, Bahrami & Latham, 2016), to the best of our knowledge, ours is the first evidence that post-decisional processing can influence motor planning within the same trial. For these reasons, we believe that our results are highly relevant to the question of how evidence accumulation and motor planning relate to each other in speeded perceptual decision-making tasks.

REVIEWERS' COMMENTS:

Reviewer #1 (Remarks to the Author):

The authors have added a new experiment (number 3) to this manuscript. In my mind, that fact alone addresses my concerns, if for no other reason as a recognition of their genuine efforts towards getting to the bottom of this issue. The question we are looking at is difficult to tackle, so there will likely be residual concerns of interpretation and potential technicalities that have not been fully exposed during peer review (see below), but I think this is inevitable. From my viewpoint, the authors have done their job. I regret that the new experiment was relegated to a supplementary Figure: I think Figure S1 should be Figure 3, and Figure 3 should be Figure 4. But I leave that choice to the authors.

Having read the comments by the other 2 reviewers (which I agree with), I suspect that they may have some residual concerns despite the addition of Experiment 3. For example, there may be a withstanding concern that the decision task is based on a binary conversion, while the motor act is more directly connected with a metric estimate, at least in terms of how they are communicated to the experimenter. Another concern may be that, despite the explicit centroid-based instructions/feedback given to participants in Exp 3, observers may have chosen to ignore said instructions and saccade to the immediately preceding targets nonetheless. I share all of those concerns, as I imagine do the authors. However, I would like to express my opinion that, although Experiment 3 may not fully address the issues at stake here, it represents a valid effort in the right direction and the overall quality of this paper should be recognized.

Reviewer #2 (Remarks to the Author):

The new experiment (Experiment 3) addressed my concerns. I recommend publication of this very interesting work.